# Chemical Composition, Enantiomeric Distribution, Antimicrobial and Antioxidant Activities of *Origanum majorana* L. Essential Oil from Nepal

**DOI:** 10.3390/molecules27186136

**Published:** 2022-09-19

**Authors:** Prem Narayan Paudel, Prabodh Satyal, Rakesh Satyal, William N. Setzer, Rajendra Gyawali

**Affiliations:** 1Department of Chemical Science and Engineering, Kathmandu University, Dhulikhel 45200, Nepal; 2Aromatic Plant Research Center, 230 N 1200 E, Suite 100, Lehi, UT 84043, USA; 3Analytica Research Center, Kirtipur 44660, Nepal; 4Department of Chemistry, The University of Alabama in Huntsville, Huntsville, AL 35899, USA; 5Department of Pharmacy, Kathmandu University, Dhulikhel 45200, Nepal

**Keywords:** *Origanum majorana* L., essential oil, bio-active component, chiral GC-MS analysis, hierarchical cluster analysis, antimicrobial activity, DPPH assay, FRAP assay

## Abstract

This study was conducted to examine the chemical constituents of *Origanum majorana* L. essential oils (EOs) that originate in Nepal, as well as their biological activities, antioxidant properties, and enantiomeric compositions. The EOs were extracted by the hydro-distillation method using a Clevenger-type apparatus and their chemical compositions were determined through gas chromatography and mass spectrometry (GC-MS). Chiral GC-MS was used to evaluate the enantiomeric compositions of EOs. The minimum inhibitory concentrations (MICs) of the essential oils were determined by the micro-broth dilution method, and the antioxidant activity was evaluated by the 2,2-diphenyl-1-picrylhydrazyl scavenging assay and ferric-reducing antioxidant power (FRAP). GC-MS analysis showed the presence of 50 and 41 compounds in the EO samples, (S_1_) and (S_2_), respectively, representing the Kathmandu and Bhaktapur districts. The oxygenated monoterpenoids, along with terpinen-4-ol, were predominant constituents in both EO samples. However, the EOs from two locations showed some variations in their major components. The chiral terpenoids for two EO samples of marjoram have also been reported in this study in an elaborative way for the first time in accordance with the literature review. A hierarchical cluster analysis based on the compositions of EOs with 50 compositions reported in the literature revealed at least 5 different chemotypes of marjoram oil. The antioxidant activity for the sample (S_2_) was found to be relatively moderate, with an IC_50_ value of 225.61 ± 0.05 μg/mL and an EC_50_ value of 372.72 ± 0.84 µg/mL, as compared to the standard used. Furthermore, with an MIC value of 78.1 µg/mL, the EO from sample (S_2_) demonstrated effective antifungal activity against *Aspergillus niger* and *Candida albicans*. Moreover, both samples displayed considerable antimicrobial activity. The results suggest that EOs of *Origanum majorana* possess some noteworthy antimicrobial properties as well as antioxidant activity, and hence can be used as a natural preservative ingredient in the food and pharmaceutical industries.

## 1. Introduction

The use of plants as traditional health remedies has become very important and popular all over the world in recent years, because more than 80% of the world’s population directly or indirectly rely on herbal drugs for their primary healthcare [1]. They have gained significant attention in recent years in fields such as medicine, nutraceuticals, dietary supplements, pharmaceutical intermediates, and chemical entities for synthetic pharmaceuticals [2,3]. The screening of such local plants for their constituents and antimicrobial properties has always been of great interest to researchers who are looking for new compounds to treat various microbial diseases [4,5].

Essential oils (EOs) are natural complex mixtures of volatile organic compounds with a strong fragrance that can be obtained from several medicinal and aromatic plants. EOs have a predominance of terpenes in which monoterpenes and sesquiterpenes are the two major classes of constituents, with isoprene as a building block of terpenes [6]. Nowadays, essential oils have a wide application in the food industry, e.g., flavors, fragrances, preservatives; in the cosmetic industry, e.g., perfumes, skin products; in feed additives, e.g., antioxidants, growth promoters, and in the pharmaceutical industry, e.g., medicines [7]. This is because they possess strong antimicrobial, antioxidant, antiparasitic, antiprotozoal, antifungal, and anti-inflammatory properties. The use of synthetic flavoring, fragrance, preservatives, and other antimicrobial compounds is growing rapidly. However, such synthetic chemicals are highly hazardous, and detrimental to health upon exceeding the permissible level of intake [8]. Therefore, essential oils may be a reasonable alternative for ensuring food safety, maintaining the nutritional content and quality of food, and removing risks to human health. Since many EOs have a predominance of oxygenated monoterpenes, which have a potential antibacterial effect, they are being used legally as flavorings [9,10,11].

*Origanum majorana* L. (syn. *Majorana hortensis* Moench), which belongs to the Lamiaceae family (Labiatae) and has medicinal values, is often known as ‘sweet marjoram’. It is a bushy perennial herb of the *Origanum* genus [12]. This herb is native to the Mediterranean region and is grown in many Asian, North African, and European nations [13]. It grows up to a height of 30 to 60 cm. It features an oblique rhizome, hairy shrub-like stalks, opposite dark green oval leaves, and clustered bracts with white or red flowers. *O. majorana* is an important culinary herb and is widely distributed in the central zone of Nepal at an altitude of about 1300 m to 3000 m, mostly in moist places. It is locally known as ‘Mu-swan’ in Newari (or maruwa phool) and ‘Raam tulsi’ in Nepali. *O. majorana* has been reported to possess very good anti-bacterial and antifungal activity against different pathogenic bacteria [14,15] and fungi [16,17]. It has also been reported to have antispasmodic, digestive, expectorant, and diuretic properties. It is effective for curing asthma and coughs and is widely used in gastronomy and natural medicine. Moreover, EOs of *O. majorana* have great potential in the cosmetic, pharmaceutical, perfume, food, and flavor industries [17,18]. The essential oil of this plant has also been used for pain, gastrointestinal problems, and respiratory tract disorders [19,20]. The antiparasitic and larvicidal activities of marjoram have also been reported in several studies [21]. Moreover, *majorana* was reported to show antidiabetic activity [22], nephrotoxicity protective effects [23], anti-inflammatory, analgesic, and anti-pyretic activities [24].

The gas chromatography and mass spectrometry (GC-MS) method has been used to investigate the mixture of bioactive volatile compounds produced by aromatic plants [25]. Similarly, an improved approach known as chiral GC-MS has been utilized in conjunction with GC-MS in order to determine the chirality of secondary metabolites. This method is critical for ensuring the integrity of the essential oil constituents, as well as assisting in the detection of impurities.

Even though there are numerous reports on the EOs of *O. majorana* from different geographical and climatic conditions of the world, there is a lack of extensive information about the chemical composition and biological activities of *O. majorana* of Nepalese origin. Because the Nepal Himalaya possesses a wide range of unique and valuable medicinal and aromatic plants (MAPs) on its land, due to its biologically diverse ecosystems, this study has set out to investigate the antimicrobial activity, chemical profiles, hierarchical cluster analysis, and antioxidant activities of EOs from two locations in Nepal and has attempted to identify the leading active constituents, along with the enantiomeric distribution of chiral terpenoids.

## 2. Results and Discussion

### 2.1. Isolation and Yields of Essential Oils

The yields of *O. majorana* essential oils from Kathmandu (sample S_1_) and Bhaktapur (sample S_2_) were found to be 0.5% (*v*/*w*) and 0.8% (*v*/*w*), respectively. The variation in the yields of EOs is attributed to various factors, such as geographical origin, harvesting period, extraction techniques, temperature, and time of extraction [26]. The essential oils of *O. majorana* were characterized by the senses, including taste, sight, smell, and touch. Both EOs of *O. majorana* were slightly viscous liquids and had a colorless to pale yellow color. They had a strong sweet and spicy odor.

### 2.2. Comparision of Chemical Composition of Two Essential Oils

The analysis of the chemical composition of *O. majorana* essential oil revealed that a total of 50 and 41 compounds were characterized in the marjoram EO for samples S_1_ and S_2_, respectively, representing 91.19% to 98.80% of total volatile oil. The relative percentages of all individual components present in the *O. majorana* EOs are shown in Table 1.

For our *O. majorana* oil (S_1_), terpinen-4-ol (32.1%), linalool (13.8%) and γ-terpinene (9.5%) were the most prominent compounds, followed by linalyl acetate (5.9%), α-terpinene (5%), *cis*-sabinene hydrate (4.4%), α-terpineol (3.7%), terpinolene (2.5%), bornyl acetate (2.4%), β-caryophyllene (2.4%), *cis*-*p*-menth-2-en-1-ol (1.8%) and *p*-cymene (1.8%) in smaller amounts. Similarly, terpinen-4-ol (33.35%), linalool (15.37%), *p*-cymene (6.90%) and linalyl acetate (6.67%) were the most major compounds, followed by *cis*-sabinene hydrate (3.48%), 1,4-hydroxy cineole (3.35%), bornyl acetate (2.83%), α-terpineol (2.63%), and caryophyllene oxide (2.54%) in smaller amounts in the oil of marjoram (S_2_). The oxygenated monoterpenoids, monoterpene hydrocarbons and sesquiterpene hydrocarbons were the major classes of terpenes in these *O. majorana* essential oils. Figure 1. shows a typical GC-MS chromatogram of *O. majorana* essential oil, displaying the separation of chemical components. Similarly, Figure 2. depicts the chemical structure of the prominent constituents in the essential oil of samples (S_1_) and (S_2_). The obtained results are in accordance with the reports of several studies carried out previously because they also found terpinen-4-ol as the most predominant compound, along with other major compounds in the marjoram EOs [15,27,28,29,30,31,32]. Table 2. shows the major components of *Origanum majorana* essential oil from different countries, which makes the comparative study easier. In a similar way, a study carried out on the EO of *majorana* from Nepal found terpinen-4-ol (22.42%), γ-terpinene (14.69%) and linalool (11.61%) as the main components [33]. Some other studies reported that terpinen-4-ol, either alone or in combination with *cis*-sabinene hydrate, linalyl acetate, γ-terpinene, etc., was found as the predominant constituent, along with some others, such as α- terpineol, α-terpinene, linalyl acetate and linalool in the *majorana* oil [23,34,35,36], which are also mentioned in Table 2. However, some other studies reported that thymol or carvacrol were the predominant compounds in marjoram EO [37]. Likewise, *O. majorana* oil from Turkey exhibited carvacrol (78.27–79.46%) as the major component [38]. In general, *O. majorana* EO is rich in terpinen-4-ol, *cis*-sabinene hydrate, γ-terpinene, α-terpinene, α-terpineol, *p*-cymene and linalool, which can be clearly observed in Table 2.

Here, the variations in the chemical content and compositions of *O. majorana* essential oil across the world might be attributed to several factors, such as the varied agro-climatic (climatical, seasonal, geographical) conditions of the regions, isolation regimes, plant species, adaptive metabolism of plants, and the plant part being analyzed [39]. Indeed, two *O. majorana* oils from Nepal (S_1_ and S_2_) showed some minor variations in volatile constituents. However, both EOs were found to be consistent with the reports presented previously, except for some slight variations due to the environmental and climatic conditions. The high content of terpinen-4-ol may be attributed to the rearrangements of components during the distillation processes, which were mentioned in the studies carried out previously, and *cis*-sabinene hydrate (responsible for the intense spicy marjoram aroma) is also present in our sample [29]. This study will provide information regarding *O. majorana* EO from Nepalese land, for those who wish to carry out further research on it.

**Table 1 molecules-27-06136-t001:** Chemical composition of essential oils of *Origanum majorana* L. from Nepal.

RI	Compound Name (S_1_)	%	RI	Compound Name (S_2_)	%
849	(3*Z*)-Hexenol ^e^	tr	**-**	**-**	**-**
920	Tricyclene ^a^	tr	**-**	**-**	**-**
923	α-Thujene ^a^	0.2	924	α –Thujene ^a^	0.05
930	α-Pinene ^a^	0.5	931	α –Pinene ^a^	0.15
947	Camphene ^a^	0.3	948	Camphene ^a^	0.09
971	Sabinene ^a^	**3.6**	972	Sabinene ^a^	1.29
975	β-Pinene ^a^	0.3	978	β –Pinene ^a^	0.12
987	Myrcene ^a^	1.3	989	Myrcene ^a^	0.16
1003	*p*-Mentha-1(7),8-diene ^b^	tr	**-**	-	**-**
1005	α-Phellandrene ^a^	0.1	**-**	-	**-**
1016	α-Terpinene ^a^	**5.0**	**-**	-	**-**
1023	*p*-Cymene ^a^	1.8	1024	*p*-Cymene ^a^	**6.90**
1027	Limonene ^a^	1.4	1028	Limonene ^a^	0.52
1029	β-Phellandrene ^a^	1.1	1029	β -Phellandrene ^a^	0.06
1030	1,8-Cineole ^b^	0.1	1031	1,8-Cineole ^b^	0.12
1033	(*Z*)-β-Ocimene ^a^	0.1	**-**	-	**-**
1042	Benzene acetaldehyde ^f^	tr	**-**	-	**-**
1043	(*E*)-β-Ocimene ^a^	0.1	**-**	-	**-**
1058	γ-Terpinene ^a^	**9.5**	**-**	-	**-**
1070	*cis*-Sabinene hydrate ^b^	**4.4**	1071	*cis*-Sabinene hydrate ^b^	**3.48**
**-**	-	**-**	1087	*trans*-Linalool oxide ^b^	0.52
1084	Terpinolene ^a^	**2.5**	**-**	-	**-**
1088	*p*-Cymenene ^a^	tr	**-**	-	**-**
1102	Linalool ^b^	**13.8**	1099	Linalool ^b^	**15.37**
1103	*trans*-Sabinene hydrate ^b^	1.6	1102	*trans*-Sabinene hydrate ^b^	0.71
1124	*cis-p*-Menth-2-en-1-ol ^b^	1.9	1124	*cis*-*p*-Menth-2-en-1-ol ^b^	1.35
**-**	-	**-**	1124	Cyclooctanone	0.47
1142	*trans-p*-Menth-2-en-1-ol ^b^	1.0	1142	*trans*-*p*-Menth-2-en-1-ol ^b^	0.78
1154	Menthone ^b^	0.1	**-**	-	**-**
1174	Borneol ^b^	0.2	**-**	-	**-**
1186	Terpinen-4-ol^b^	**32.1**	1180	Terpinen-4-ol ^b^	**33.35**
**-**	-	**-**	1187	*p*-Cymen-8-ol ^b^	0.53
**-**	-	**-**	1188	3-*cis*-Hexenyl butyrate ^b^	0.13
**-**	-	**-**	1190	1,4-Hydroxy cineole ^b^	**3.35**
1196	α-Terpineol ^b^	**3.7**	1195	α –Terpineol ^b^	**2.63**
1197	*cis*-Piperitol ^b^	0.4	1197	*cis*-Piperitol ^b^	0.12
1203	*p*-Cumenol ^g^	tr	-	-	-
1208	*trans*-Piperitol ^b^	0.6	1208	*trans*-Piperitol ^b^	0.26
**-**	-	**-**	1224	Isoascaridole ^b^	0.15
1214	*cis*-Sabinene hydrateacetate ^b^	tr	**-**	-	**-**
1222	Nerol ^b^	0.2	**-**	-	**-**
1236	Pulegone ^b^	tr	**-**	-	**-**
1248	Linalyl acetate ^b^	**5.9**	1252	Linalyl acetate ^b^	**6.67**
-	-	-	1255	*p*-menthane-1,2,3-triol ^b^	0.69
1273	*trans*-Ascaridol glycol ^b^	0.1	1276	*trans*-Ascaridol glycol ^b^	1.15
1282	Bornyl acetate ^b^	**2.4**	1282	Bornyl acetate ^b^	**2.83**
**-**	-	**-**	1291	Terpinen-4-ol acetate ^b^	0.31
**-**	-	**-**	1297	Carvacrol ^b^	0.08
**-**	-	**-**	1345	2-Methyl-2-(para-tolyl) propionaldehyde	0.24
**-**	-	**-**	1354	Terpen-diol	0.46
1293	Terpin-1-en-4-yl acetate ^b^	0.2	**-**	-	**-**
1329	δ-Elemene ^c^	tr	**-**	-	**-**
1355	Neryl acetate ^b^	0.2	1361	Neryl acetate ^b^	0.14
1375	Geranyl acetate ^b^	0.3	1377	Geranyl acetate ^b^	0.48
1417	β-Caryophyllene ^c^	**2.4**	1418	β –Caryophyllene ^c^	0.13
**-**	-	**-**	1486	Hydroxy linalyl acetate ^b^	1.12
**-**	-	**-**	1488	*p*-Menthane-1,2,4-triol	1.44
1452	α-Humulene ^c^	0.1	**-**	-	**-**
1492	Bicyclogermacrene ^c^	0.4	**-**	-	**-**
1500	(*E*,*E*)-α-Farnesene ^c^	tr	**-**	-	**-**
1573	Spathulenol ^d^	tr	1579	Spathulenol ^d^	0.14
1578	Caryophyllene oxide ^d^	0.1	1580	Caryophyllene oxide ^d^	**2.54**
**-**	-	**-**	1612	Humulene epoxide II ^d^	0.13
2047	Abietatriene ^h^	tr	**-**	-	**-**
	Total identified	**99.8**			**91.19**

Note: RI = retention index determined in reference to a series of *n*-alkanes (C8–C40) on a ZB-5ms column; compounds are listed in order of elution (increasing RI). tr = trace (˂0.05%), ‘-‘ = not detected;% = percent composition. Symbols: ‘a’= monoterpene hydrocarbon; ‘b’ = oxygenated monoterpene; ‘c’ = sesquiterpene hydrocarbon; ‘d’ = oxygenated sesquiterpene; ‘e’ = alcohol; ‘f’ = aldehyde; ‘g’ = phenol carrying an isopropyl group at position 4; and ‘h’ = diterpenoid. “Sample S_1_”= Kathmandu district; “Sample S_2_”= Bhaktapur district.

**Table 2 molecules-27-06136-t002:** The major chemical compounds in the essential oil of *O. majorana* in the different countries.

Country	Major compounds	References
Morocco	Terpinen-4-ol (34.1%), α-Terpinene (19.2%), Terpineol (8.9%)	[15]
Iran	Terpinen-4-ol (32.69%), γ -Terpinene (12.88%), trans-Sabinene hydrate (8.47%), α -Terpinene (7.98%)	[27]
Iran	Terpinene-4-ol (22.15%-25.65%), γ-Terpinene (13.94%-16.22%), α-Terpinene (8.11%-10.39%), α-Terpineol (4.53%-6.39%)	[28]
India	Terpinen-4-ol (31.15%), *cis*-Sabinene hydrate (15.76%), *p*-Cymene (6.83%), Sabinene (6.91%), trans-Sabinene hydrate (3.86%), α-Terpineol (3.71%)	[29]
Tunisia	Terpinene-4-ol, γ-Terpinene, *cis*-Sabinene-hydrate, α-Terpineol	[30]
China	Terpinen-4-ol (33.0 %), Caryophyllene oxide (11.9 %), *p*-Cymene (6.8 %), α-Terpineol (6.7 %), Spathulenol (6.0 %)	[31]
Egypt	Terpinen-4-ol (30.4%), γ-Terpinene, *cis*-Sabinene hydrate, α-Terpinene, Sabinene, α-Terpineol	[32]
Reunion Island	Terpinen-4-ol (38.4%), *cis*-Sabinene hydrate (15.0%), *p*-Cymene (7.0%), γ-Terpinene (6.9%).	[40]
Albania	Terpinen-4-ol (21.3%), *trans*-Sabinene hydrate (15.5%), γ-Terpinene (14.0%), α-Terpinene (8.9%)	[41]
Tunisia	Terpinen-4-ol	[4]
Hungary	Terpinen-4-ol	[11]
Tunisia	Terpinen-4-ol (23.2%), cis-Sabinene hydrate (17.5%), γ-Terpinene (10.5%), *p*-Cymene (9%), α–Terpineol (5.6%), α-Terpinene (4.7%), trans-Sabinene hydrate (4.0%)	[14]
Nepal	Terpinen-4-ol (22.42%), Linalool (11.61%), γ-Terpinene (14.69%), α-Terpineol (7.02%), α-Phellandrene (9.8%), *p*-Cymene (8.91%)	[33]
Egypt	*cis*-Sabinene hydrate, Linalyl acetate, 𝛾-Terpinene	[23]
Cyprus	cis-Sabinene hydrate (7.4–33.3%), Terpinen- 4-01 (16.6–21.6%), α-Terpineol (7.3%), trans-Sabinene hydrate, (4.7%), γ-Terpinene (8.3%), α-Terpinene(4.7%)	[34]
Turkey	Trace amounts of Carvacrol. *cis*-Sabinene hydrate (30–44%), Terpinen-4-ol (8–14%)	[36]
Greece	4-Terpineol (37%), *p*-Cymene (12%), α-Terpineol (7%)	[42]
Iran	Linalyl acetate (26.1%), Sabinene (12%)	[43]
Brazil	𝛾-Terpinene (25.73%), 𝛼-Terpinene (17.35%), Terpinen-4-ol (17.24%), Sabinene (10.8%)	[44]
Venezuela	*cis*-Sabinene hydrate (30.2%), Terpinen-4-ol (28.8%), γ-Terpinene (7.2%), α-Terpineol (6.9%), *trans*-Sabinene hydrate (4.4%), Linalyl acetate (3.8%), α-Terpinene (3.6%)	[45]

### 2.3. Chiral GC-MS Analysis for Enantiomeric Distribution

The chiral GC-MS analysis was performed for the identification of enantiomeric compounds that are present in the EOs of *O. majorana*. The relative percentages of the dextrorotatory and laevorotatory compounds identified in the *O. majorana* EO are presented in Table 3. It also shows the enantiomeric compositions of *majorana* oil and identified the presence of twelve chiral compounds in total for both EOs. Figure 2 shows the structure of some of the chiral compounds present in both EO samples. The determination of enantiomeric composition is a powerful tool in order to authenticate the EOs because EOs obtained from different plants may be adulterated, due to the addition of several foreign components. In simple terms, geographical location and distillation time do not affect the enantiomeric distribution of chiral compounds present in the EOs. This analysis is the first attempt to capture the marjoram EO in more detail, which belong to Nepalese origin. Terpinen-4-ol is the major oxygenated monoterpene found in the EO of our two samples, in which the dextrorotatory (+) enantiomer is the most predominant chiral compound. In our study, the essential oil of sample (S_2_) has nearly the racemic mixture of terpinen-4-ol, (+) 52.6% and (–) 47.4%, whereas terpinen-4-ol, (+) 58.31%, is dominant over terpinen-4-ol, (–) 47.69%, for the EO for sample (S_1_). In both EO samples, (–) β-caryophyllene, (–) bornyl acetate, and (–) linalyl acetate were detected as enantiomerically pure with 100% in levorotatory form. Camphene is the enantiomerically most dominant component that exists in levorotatory form with slight variations, including (–) 96.8% to (–) 97.87% in our both EO samples, which is followed by (+) sabinene and (+) *cis*-sabinene hydrate, with the variation in the enantiomeric distribution in this study. Terpinen-4-ol in the EO of majorana from Israel was reported as not optically pure and the enantiomeric composition was about (+)-terpinen-4-ol (73.0%) and (–)-terpinen-4-ol (27.0%) [46]. The enantiomeric distribution of linalool was reported only in a sample of marjoram oil with (–)-linalool (82.0%) and (+)-linalool (18.0%) [47]. These previous results were found to be in close agreement with the chiral distribution of linalool and terpinen-4-ol as compared to our EOs, although we did not obtain information from the previous study on the enantiomeric distribution for other components in detail. Finally, we can conclude that the (+)/(−) ratios of each of the terpenoids remain relatively constant in our samples, regardless of the geographical location of the oil source.

### 2.4. Chemotypes of O. majorana EOs

In order to highlight the chemotypes of *O. majorana* EOs, we performed an agglomerative hierarchical cluster (AHC) analysis based on the chemical compositions of the two EOs (S_1_ and S_2_) under this study, along with fifty additional marjoram oil chemical compositions from the literature review. The dendrogram of this analysis is shown in Figure 3. From the analysis of AHC, there are five different chemotypes, which are classified as follows: (1) terpinen-4-ol/γ-terpinene/α-terpinene/sabinen/α-terpineol, (2) *cis*-sabinene hydrate/terpinen-4-ol/γ-terpinene, (3) *trans*-sabinene hydrate/*cis*-sabinene hydrate/terpinen-4-ol/α-terpineol, (4) linalool/*p*-cymene/estragole and (5) carvacrol/linalool/*p*-cymene.

Chemotype-1, dominated by terpinen-4-ol [16,28,31,32,33,37,48,49,50], is a cluster made up of twenty-five samples, including both samples (S_1_ and S_2_) from Nepal in our study, along with the previous marjoram sample from Nepal-1. Chemotype-2, dominated by *cis*-sabinene hydrate [29,36,39,45,51,52], has eight samples. Chemotype-3, dominated by *trans*-sabinene hydrate [53,54,55,56,57,58], has twelve samples in this analysis. Chemotype-4 is dominated by linalool [59,60] and is comprised of three samples. The chemotype-5 is dominated by carvacrol, which is comprised of four samples [38,61,62,63]. 

### 2.5. Antibacterial and Antifungal Activity

The antimicrobial activities of *O. majorana* essential oils were examined against several microorganisms. Both EOs were noted to be active against all microbial strains, but their efficacies were found to be variable to a different degree. The results obtained for the antimicrobial activities are given in Table 4. In the EO of marjoram for sample (S_1_), terpinen-4-ol, linalool and γ-terpinene were the major components. The essential oil sample (S_2_) had terpinen-4-ol, linalool and *p*-Cymene as the most dominant components. The EO for sample (S_2_) showed moderate antibacterial activity against *Staphylococcus aureus,* with an MIC value of 156 µg/mL. Both EOs had weaker activity against *Bacillus cereus* and *Staphylococcus epidermidis,* with an MIC value of 312.5 µg/mL as compared to the positive control, gentamicin (MIC = 19.5 µg/mL). The EO for sample (S_2_) exhibited good antifungal activity against *Aspergillus niger* and *Candida albicans,* with an MIC value of 78.1 µg/mL. Similarly, it showed moderate antifungal activity against *Trichophyton mentagrophytes and Aspergillus fumigatus,* with an MIC value of 156 µg/mL. The EO for sample (S_1_) had moderate antifungal activity against *Aspergillus niger, Candida albicans* and *Trichophyton mentagrophytes.* However, when compared to the positive control, amphotericin B (MIC = 19.5 µg/mL), both EOs were found to have weaker activity against the remaining fungal strains, with an MIC value of 312.5 µg/mL.

According to several investigations, the EO of majoram showed a moderately varied antibacterial effect on *Staphylococcus aureus*, with an MIC value of 782 µg/mL [32], 192 μg/mL [14], 50 μg/mL [64], from 150.0 to 250.0 μg/mL [65], 50 μg/mL [64], and from 2.5 μL/mL (for *S. aureus* BH3), 5 μL/mL(for *S. aureus* BH01) to 10 μL/mL(for *S. aureus* BH02), respectively [66]. Similarly, this oil had a good antibacterial effect on *Bacillus cereus*, with an MIC value of 75.0 to 150.0 μg/mL [65] and 97 μg/mL [14]. It had an antibacterial effect on *Staphylococcus epidermidis*, with an MIC value of 390 µg/mL [14]. The marjoram EOs also exhibited strong effects against different fungal species in terms of MICs or growth inhibition doses. Likewise, strong inhibitory effects were observed with 10 μg/mL at 90% concentration [67], 2000 ppm concentration [16] and 22 mg/mL concentration [68] against *Aspergillus niger*. The *majorana* EO showed very good antifungal activity against *Candida albicans,* with MIC values of 58 µg/mL and 468 µg/mL [14]. Similarly, against *Microsporum canis* and *Trichophyton rubrum*, marjoram EO had mild activity with MIC values of 234 µg/mL, while it had moderate activity against *Trichophyton mentagrophytes,* with an MIC value of 117 µg/mL [14]. On the basis of the literature reviews, we can conclude that these results obtained for antimicrobial activities of EOs are in close agreement with some of the previous reports in the literature.

The antimicrobial activities of the EOs are mostly associated with their high content of oxygenated monoterpenes, particularly major components, such as terpinen-4-ol [69], α-terpineol, α-pinene, *p*-cymene etc. γ-terpinene, β-caryophyllene and sabinene are also known for their strong antimicrobial activities [70]. Indeed, the observed activity of EOs against bacterial and fungal species may be attributed to the synergistic effect among them and other constituents, rather than a single constituent. Our finding is in agreement with the previous studies that the essential oils exhibit more antifungal substances from aromatic plants than antibacterial substances, especially in the Lamiaceae [15,16]. This result may indicate that the essential oil of *O. majorana* can be used as natural preservatives in food against foodborne diseases and food spoilage, such as *Bacillus* sp., *Staphylococcus aureus*, *Candida* sp., *Fusarium* sp., *Aspergillus* sp., etc. 

### 2.6. Antioxidant Activity

#### 2.6.1. DPPH free Radical Scavenging Activity

In the present study, the antioxidant activity of *O. majorana* EO was determined by using a 2,2-diphenyl-1-picrylhydrazyl (DPPH) radical scavenging method and was compared to ascorbic acid activity. The antioxidant activity of EO samples is summarized in Table 5 in terms of IC_50_ values.

The DPPH is a stable free radical, which accepts the hydrogen radical or an electron, forming a stable diamagnetic molecule [71]. Indeed, DPPH solution is initially purple in color, which fades into the yellow color of diphenylpicryl hydrazine when bioactive components (antioxidants) from essential oils donate hydrogen radicals [72]. A standard criterion for measuring the antioxidant activity of EO samples is the IC_50_ values, which represents the concentration of antioxidant needed to minimize the initial DPPH concentration by 50% [72]. Our study revealed that *O. majorana* EO from two locations exhibited moderate DPPH free radical-scavenging activity, with IC_50_ values of 503.08 ± 0.06 μg/mL (S_1_) and 225.61 ± 0.05 μg/mL (S_2_), as compared to the results reported previously. Here, marjoram EO (S_2_) showed slightly stronger antioxidant activity than that of sample (S_1_). However, the antioxidant activities of both EO samples were lower than that of the positive control, ascorbic acid (9.74 ± 0.07 μg/mL). The average percentage of free-radical scavenging activity of essential oils and the reference standard (ascorbic acid) by DPPH assay is shown in Figure 4a,b. According to a literature review, *O. majorana* EOs were found to exhibit stronger antioxidant activity in terms of smaller IC_50_ values, smaller than 200 μg/mL [14,21,63,73,74,75]. In other studies, antioxidant activity of marjoram EO was reported to be weaker, with IC_50_ values greater than 200 μg/mL [41,68,76,77]. Based on these previous reports, both our EO samples may be considered moderate antioxidant sources.

Our EO observed radical-scavenging activity could be entirely attributed to the higher activity of the major component terpinen-4-ol. Several other EO components have also exhibited DPPH radical-scavenging action, despite the fact that phenolic compounds are typically recognized as being the source of the strongest antioxidant activity. However, it is hard to attribute the radical scavenging activity to one or a few active volatile compounds of the total EO, because EO is a complex mixture of different volatile compounds. Generally, the antioxidant activity of the whole essential oil showed better radical scavenging capacity than the individual components, indicating the possible synergistic interaction between different components of essential oils [75].

#### 2.6.2. Ferric-Reducing Antioxidant Power (FRAP)

For further confirmation of the antioxidant activity of EO samples, we investigated the total reducing power of EOs using the FRAP assay. The results showed that there is an increase in absorbance at 700 nm with an increase in the concentration of EO samples. This is because the assay involves the transformation of Fe^3+^/ferricyanide complexes to the ferrous (Fe^2+^) form in the presence of reducers (i.e., antioxidants) in EO extracts. The results are expressed in terms of effective concentration (EC_50_), which are listed in Table 5 and Figure 5. In general, the lower the EC_50_ values, the higher the reducing ability of the EO extract to convert ferric to the ferrous ion form.

From the analysis, it was observed that the EO of sample (S_2_) has higher reducing ability, with an EC_50_ value of 372.72 ± 0.84 µg/mL, than that of sample (S_1_), with an EC_50_ value of 511.4 3 ± 0.61 µg/mL. On the other hand, both EO samples showed lower EC_50_ values as compared to the standard positive control, ascorbic acid, at 217.23 ± 0.34 µg/mL. However, the EO sample (S_1_) showed considerable reducing power. This reducing capacity could presumably be due to their hydrogen donating ability from the phenolic compounds [78]. In addition to this, the number and position of the hydroxyl groups of phenolic compounds also lead to their antioxidant activity [79].

## 3. Materials and Methods

### 3.1. Plant Material Collection

The fresh plant samples of *O. majorana* during the flowering stage were collected in May/June 2019, from the following two locations: Nagarjun, Kathmandu with latitude 27°43′58.857″ N and longitude 85°15′26.5572″ E at an elevation of 1537 m, (S_1_) and Sanothimi, Bhaktapur with latitude 27°40′48.8856″ N and longitude 85°22′42.3654″ E at an elevation of 1336 m (S_2_) in Nepal. Figure 6a,b show the geographical locations for plant sample collection and a photograph of the plant sample. The taxonomic identification of plant materials was confirmed by Ms. Rita Chhetri (senior research officer), at the National Herbarium and Plant Laboratories, Godawari, Lalitpur, Nepal.

### 3.2. Extraction of Essential Oils

The shade-dried and chopped aerial parts of plant material (ca. 100 g) were submitted to hydro-distillation for 3 hours with 500 mL of distilled water, using a Clevenger-type apparatus (Jain Scientific Glass Works, JSGW, India) following standard protocol [80]. The obtained essential oil was dried over anhydrous sodium sulfate and, after filtration, stored at 4 °C until further testing and analysis. The yields were calculated based on the volume to weight ratio with the percentages 0.5% (S_1_) and 0.8% (S_2_) of EO samples of *O. majorana*.

### 3.3. Chemical Composition Analysis by Gas Chromatography-Mass Spectrometry

The essential oil of *O. majorana* was analyzed by GC-MS using a Shimadzu GC-MS-QP2010 Ultra apparatus (Shimadzu Scientific Instruments, Columbia, MD, USA) operated in the electron impact (EI) mode (electron energy = 70 eV), with a scan range of 40–400 amu, scan rate of 3.0 scans/s, and GC-MS solution software version 4.5 (Shimadzu Scientific Instruments, Columbia, MD, USA). The GC column was a ZB-5 MS fused silica capillary column with a (5%phenyl)–polymethyl siloxane stationary phase and a film thickness of 0.25 μm. The carrier gas was helium, with a column head pressure of 552 kPa and a flow rate of 1.37 mL/min. The injector temperature was 250 °C and the ion source temperature was 200 °C. The GC oven temperature program was programmed with 50 °C initial temperature and the temperature increased at a rate of 2 °C/min to 260 °C. A 5% *w*/*v* solution of the sample in CH_2_Cl_2_ was prepared and 0.1 μL was injected with a splitting mode (30:1) [81,82].

Identification of the oil components was based on their retention indices, determined by reference to a homologous series of *n*-alkanes (C8–C40), and by comparison of their mass spectral fragmentation patterns with those found in the MS databases using LabSolution GC-MS solution software and those reported in the literature [83]. The relative percentages of the individual components are listed in Table 1.

### 3.4. Chiral GC-MS analysis for Enantiomeric Components

Enantiomeric analysis for *O. majorana* oil was carried out by using a Shimadzu GC–MS–QP2010S (Shimadzu Scientific Instruments, Columbia, MD, USA) with EI mode (70 eV) and a B-Dex 325 chiral capillary GC column. It allows scanning at a rate of 3.0 scans/s in the range of 40–400 m/z. The column temperature was set at 50 °C, then increased by 1.5 °C/min until it reached 120 °C, and then by 2 °C/min until it reached 200 °C. The final temperature of the column was 200 °C and it was kept constant. Helium was used as the carrier gas, with a constant flow rate of 1.8 mL/min. A 3% *w*/*v* solution in CH_2_Cl_2_ was prepared for the essential oil sample, and 0.1 µL was injected at a split ratio of 1:45 [44,45,47]. The percentage composition of enantiomers was calculated from the peak area. The enantiomers were identified by comparing retention times and mass spectral fragmentation patterns with authentic samples acquired from Sigma-Aldrich (Milwaukee, WI, USA). The enantiomeric distribution of components for EO is presented in Table 3.

### 3.5. Hierarchical Cluster Analysis for Chemical Composition of EOs

A total of 50 *O. majorana* essential oil compositions from the different published literatures, besides the 2 EO samples (S_1_ and S_2_) from this study, were taken as the operational taxonomic units (OTUs). The percentage composition of major essential oil components (such as terpinen-4-ol, γ-terpinene, *cis*-sabinene hydrate, linalool, *trans*-sabinene hydrate, carvacrol, α-terpinene, α-terpineol, sabinene, *p*-cymene, linalyl acetate, β-caryophyllene, α-terpinolene, myecene, pulegone, α-pinene, *p*-menth-1-en-4-ol, thymol, *cis*-sabinene hydrate acetate, β-phellandrene, limone, *cis*-*p*-Menth-2-en-1-ol, bicyclogermacrene, camphene, α- phellandrene, 2-carene, β –pinene, estragole, α-thujene, caryophyllene oxide, *trans*-4 thujanol, *trans*-*p*-Menth-2-en-1-ol, and bornyl acetate) was used to determine the chemical relationship between the various *O. majorana* essential oil samples by agglomerative hierarchical cluster (AHC) analysis, using the IBM SPSS STATISTICS VERSION 8.5.5, IBM: Armonk, NY, USA.

### 3.6. Antimicrobial Activity

The in-vitro antimicrobial activities of essential oils were evaluated in terms of minimum inhibitory concentration (MIC) using the micro-broth dilution technique. The bacterial strains used were *Bacillus cereus* (ATCC 14579), *Staphylococcus aureus* (ATCC 29213), and *Staphylococcus epidermidis* (ATCC 14990). All bacteria were cultured on tryptic soy agar (Sigma-Aldrich, St. Louis, MO, USA). The solution of each essential oil was prepared at a concentration of 5000-µg/mL, using dimethyl sulfoxide (DMSO). Then, 50 µL of this solution was diluted with 50 μL of cation-adjusted Mueller Hinton broth (CAMHB) (Sigma-Aldrich, St. Louis, MO, USA), which was transferred to the top well of a 96-well micro-dilution plate. The solution of the EO sample thus prepared was serially two-fold diluted in fresh CAMHB, resulting in the final concentrations of 2500, 1250, 625, 312.5, 156.3, 78.1, 39.1 and 19.5 µg/mL. The bacterial strains were obtained from a fresh culture and added to each well of 96-well micro-dilution plates (~ 0.1 μL) at a concentration of approximately 1.5 × 10^8^ CFUs/mL (determined using Mcfarland standard). Then, they were incubated at 37 °C for 24 hours. Gentamicin (Sigma-Aldrich, St. Louis, MO, USA) was used as a positive antibiotic control, and DMSO was used as a negative control (50 µL DMSO diluted in 50 µL broth medium and then serially diluted as mentioned above). This experiment was carried out using the standard protocols [84,85,86].

The fungal strains were Aspergillus niger (ATCC 16888), Candida albicans (ATCC 18804), Trichophyton mentagrophytes (ATCC 18748), Aspergillus fumigatus (ATCC 96918), Cryptococcus neoformans (ATCC 32045), Microsporum canis (ATCC 11621), Microsporum gypseum (ATCC 24102), and Trichophyton rubrum (ATCC 28188). All fungi were cultured on yeast-nitrogen base growth medium (Sigma-Aldrich, St. Louis, MO, USA). All the stock solutions of EOs were prepared in accordance with the method discussed above. The freshly cultivated fungi strains, with approximately 7.5 × 10^7^ CFUs/mL final concentrations, were added to each well of 96-well micro-dilution plates, which were then incubated at 35 °C for 24 hours. Here, DMSO was used as a negative control, and amphotericin B (Sigma-Aldrich, St. Louis, MO, USA) was used as a positive control [45,47]. The antibacterial and antifungal activities (MICs) of EOs are listed in Table 4. The microorganisms were purchased from ATCC (Lines 199–203), and cells were harvested from freshly cultured plates for the further assay.

### 3.7. Antioxidant Activity

#### 3.7.1. DPPH Radical Scavenging Activity

The antioxidant activity of the EO was evaluated based on its ability to scavenge 2,2-diphenyl-1-picrylhydrazylstable free radicals (DPPH) (Sigma-Aldrich, St. Louis, MO, USA). The assay was carried out by using the colorimetric method with some modifications using the standard method [87]. Briefly, 2 mL of different concentrations of the EO were mixed with 2mL of DPPH solutions (100 µM). The mixture was then allowed to stand at room temperature in the dark for 30 minutes to complete the reaction and the absorbance was measured at 517 nm using a UV spectrophotometer (UV-1800 Shimadzu, Japan). Ascorbic acid was used as the reference standard and methanol (2mL each) was used as a negative control. The radical scavenging activity was calculated as a percentage of DPPH discoloration using the following equation:DPPH scavenging effect (%) = [(A_0_ − As)/A_0_] ∗ 100(1)
A_0_ is the absorbance of the control reaction (containing all reagents without the test sample) and A_S_ is the absorbance of the test sample. The antiradical activity was evaluated in terms of IC_50_ values (µg/mL), indicating the EO extract doses required to cause 50% inhibition. A lower IC_50_ value corresponds to higher antioxidant activity of the EO extract. A similar process was carried out for the reference standard, ascorbic acid. The experiments were conducted in triplicate. The IC_50_ values calculated for EOs and standards are listed in Table 5.

#### 3.7.2. Ferric-Reducing Antioxidant Power (FRAP)

The ferric-reducing antioxidant power (FRAP) of EO samples was analyzed following the standard method [88]. All the chemicals used in this assay were purchased from the Fisher Scientific Co., Ltd., Bengalaru, India. First, 1 mL of each of the sample concentrations was mixed with 2.5 mL of phosphate buffer (0.2 M, pH 6.6) and 2.5 mL of 1% potassium ferricyanide, K_3_[Fe(CN)_6_]. The mixture was incubated at 50° C for 20 min. After incubation, 2.5 mL of 10% trichloroacetic acid (TCA) was added to the solution, which was then centrifuged for 10 min at 1000 rpm. The supernatant was collected, and 2.5 mL of the supernatant was mixed with distilled water (2.5 mL) and FeCl_3_ (0.5 mL, 0.1%). The mixture was shaken vigorously and allowed to stand at room temperature for 30 min, and the absorbance was measured at 700 nm on a UV–visible spectrophotometer, where higher absorbance indicates higher reducing power. The above assays were carried out in triplicate, and the results were expressed as mean values ± standard deviation. Finally, the mean of absorbance values was plotted against the concentration values. Increased absorbance of the reaction mixture indicated the increased reducing power. The results were expressed as effective concentrations (EC_50_ value in μg/mL) when the absorbance was 0.5 at 700 nm and compared with standard ascorbic acid, which was used as a positive control. The EC_50_ values calculated for EOs and standards are listed in Table 5. 

### 3.8. Data Analysis

The data obtained from the overall experiment were processed and analyzed using various software, including Microsoft Excel OriginPro 2016 64Bit (Origin version 9.3, OriginLab Corporation, ORIGIN: Northampton, MA, USA) and IBM SPSS STATISTICS VERSION 8.5.5, IBM: Armonk, NY, USA, and GIS software (ArcGIS software, ESRI, Redlands, CA, USA), which was used to create a GIS map of the sampling sites. Finally, the antioxidant activity of EOs was expressed as the mean ± standard deviation (SD) of the three replications.

## 4. Conclusions

In summary, the chemical composition, biological activities, and enantiomeric distribution of *O. majorana* EOs from Nepal have been broadly determined for the first time. Our results demonstrated that EOs of *O. majorana* from two locations in Nepal were found to contain oxygenated monoterpenoids as their predominant bioactive constituents, in which terpinen-4-ol was the most dominant compound, with a higher percentage compared to previous results and with a slight variation in the major components. The *O. majorana* EO possessed moderate activity against the tested bacteria, while it showed prominent activity against the fungal strains. Likewise, the EO also showed relatively moderate antioxidant activity as compared to the studies carried out previously, although their radical scavenging activity was lower than the control. This study also reported the chiral compounds found in the EOs of marjoram, which are very useful for identifying the adulteration and authentication of samples. On the basis of the relative concentration of major components in marjoram EOs, this study showed at least five different chemotypes. Furthermore, the *O. majorana* EOs could be a promising natural ingredient in flavoring, perfumery, aromatherapy, and pharmaceuticals, due to their possession of biological properties with some synergistic and antagonistic effects that are associated with minor and/or major bioactive volatile compounds. Some in-vivo and clinical tests are needed to further justify the potency of marjoram EOs, along with the mode of action of the bioactive components present.

## Figures and Tables

**Figure 1 molecules-27-06136-f001:**
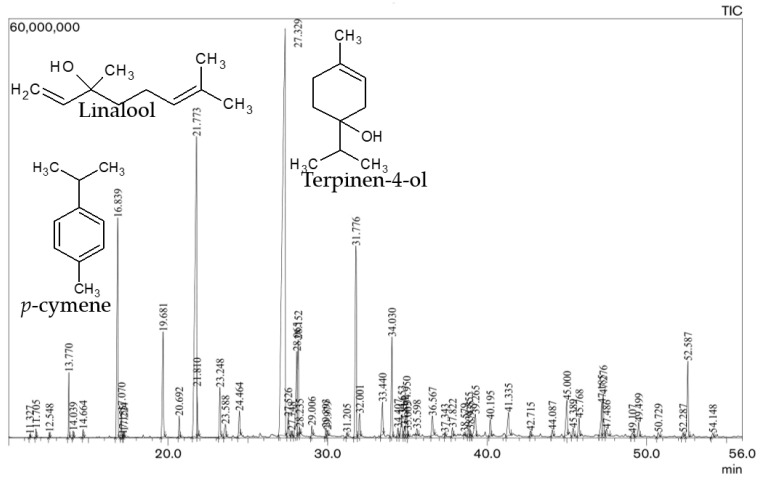
Typical GC-MS chromatogram of *O. majorana* L. essential oil.

**Figure 2 molecules-27-06136-f002:**
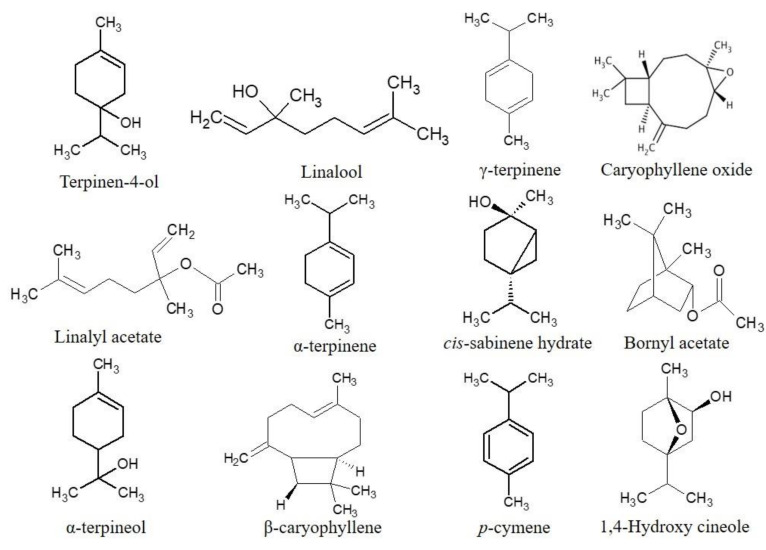
Major chemical constituents identified in the essential oil of *Origanum majorana* L. (Sample S_1_ and Sample S_2_) and the chiral compounds present, such as terpinen-4-ol, *cis*-sabinene hydrate, linalool, linalyl acetate, bornyl acetate, α-terpineol and β-caryophyllene in both EOs.

**Figure 3 molecules-27-06136-f003:**
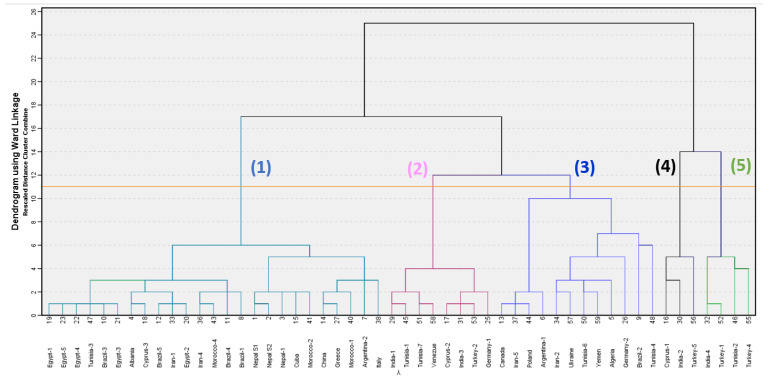
A dendrogram obtained from the agglomerative hierarchical cluster analysis of 52 *Origanum majorana* essential oil compositions. Numbers, (1), (2), (3), (4) and (5) represent different chemotypes dominated by terpinen-4-ol, *cis*-sabinene hydrate, *trans*-sabinene hydrate, linalool and carvacrol, respectively.

**Figure 4 molecules-27-06136-f004:**
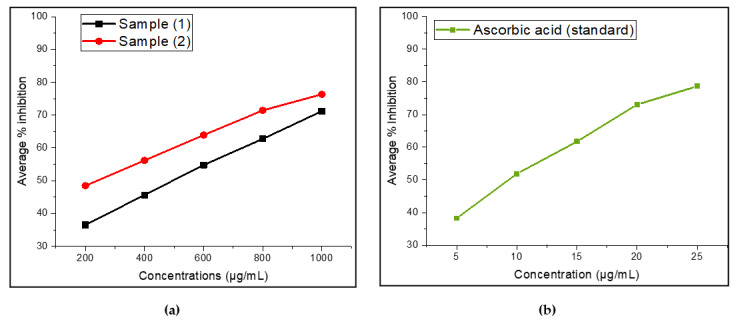
Average percentage of free-radical scavenging activity by DPPH assay (*n* = 3). (**a**) Essential oils from majorana species (S_1_ and S_2_); (**b**) standard reference (ascorbic acid).

**Figure 5 molecules-27-06136-f005:**
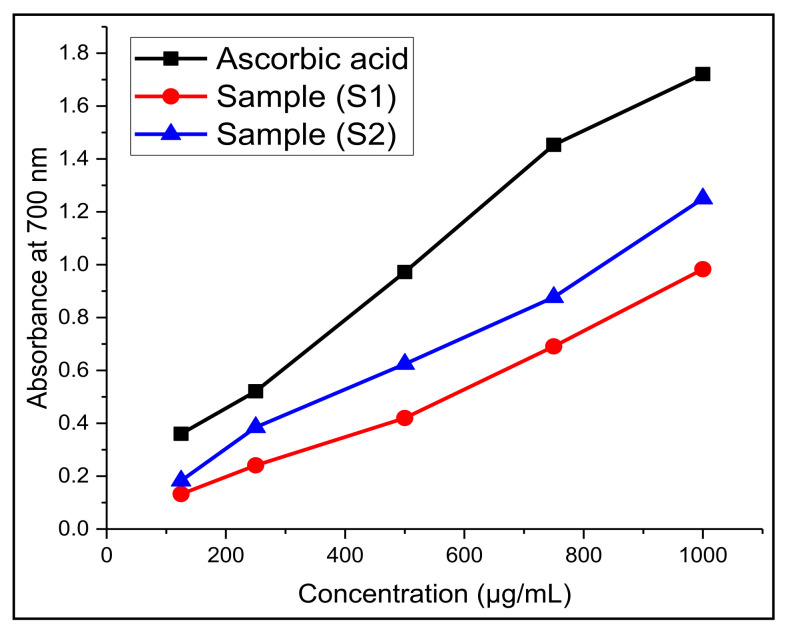
Ferric-reducing antioxidant power of O. majorana EO (S_1_ and S_2_) and standard reference (ascorbic acid) (*n* = 3).

**Figure 6 molecules-27-06136-f006:**
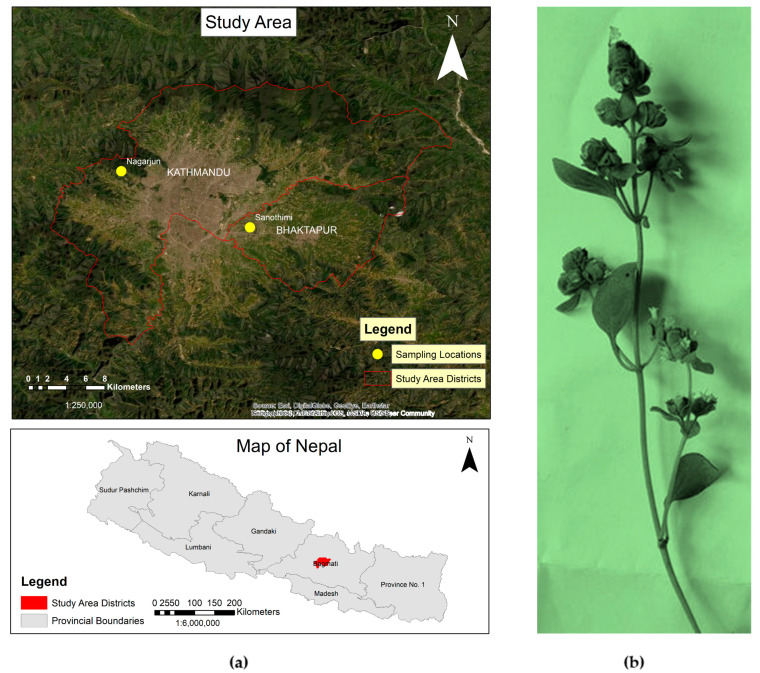
(**a**) The geographical location of *majorana* species collection sites; (**b**) a photograph of a twig of *O. majorana* L.

**Table 3 molecules-27-06136-t003:** Enantiomeric distributions of chiral compounds in EOs of *Origanum majorana* L. from Nepal.

Compounds	*O. majorana* (Sample S_1_)		*O. majorana* (Sample S_2_)	
	+ (D)	– (L)	+ (D)	– (L)
α-Pinene	55.81	44.19	50.81	49.2
Camphene	2.31	97.87	3.18	96.8
Sabinene	95.63	4.37	91.6	8.4
β-Pinene	24.31	75.69	29.3	70.7
Limonene	66.48	33.52	66.82	33.2
*cis*-Sabinene hydrate	88.52	11.48	86.8	13.2
Linalool	31.43	68.57	29.6	70.4
Terpinen-4-ol	58.31	41.69	52.6	47.4
Linalyl acetate	0.0	100.0	0.0	100.0
Bornyl acetate	0.0	100.0	0.0	100.0
α-Terpineol	75.3	24.7	72.4	27.6
β-Caryophyllene	0.0	100.0	0.0	100.0

**Table 4 molecules-27-06136-t004:** Minimum inhibitory concentrations (MICs) of *O. majorana* essential oils against tested bacterial and fungal strains.

Name of Micro-Organism	MICs (µg/mL)	
	EO Sample (S_1_)	EO Sample (S_2_)
*Bacillus cereus* (ATCC 14579)	312.5	312.5
*Staphylococcus aureus* (ATCC 29213)	312.5	156.3
*Staphylococcus epidermidis* (ATCC 14990)	312.5	312.5
*Aspergillusniger* (ATCC 16888)	156.3	78.1
*Candida albicans* (ATCC 18804)	156.3	78.1
*Trichophytonmentagrophytes* (ATCC 18748)	156.3	156.3
*Aspergillusfumigatus* (ATCC 96918)	312.5	156.3
*Cryptococcus neoformans* (ATCC32045)	312.5	312.5
*Microsporumcanis* (ATCC11621)	312.5	312.5
*Microsporumgypseum* (ATCC24102)	312.5	312.5
*Trichophytonrubrum* (ATCC28188)	312.5	312.5

Note: Gentamicin was used as the standard for bacteria (MIC = 19.5 µg/mL) and amphotericin B as the standard for fungi (MIC = 19.5 µg/mL). MICs = minimum inhibitory concentrations (in µg/mL).

**Table 5 molecules-27-06136-t005:** Antioxidant activity of *O. majorna* EOs (S_1_ and S_2_) and ascorbic acid (standard).

Samples and Standard	DPPH Radical Scavenging IC_50_ Value (µg/mL)	FRAP EC_50_ Value (µg/mL)
*O. majorana* (S_1_)	503.08 ± 0.06	511.43 ± 0.61
*O. majorana* (S_2_)	225.61 ± 0.05	372.72 ± 0.84
Ascorbic acid	9.74 ± 0.07	217.23 ± 0.34

Note: Values are mean ± standard deviations from three experiments (*n* = 3).

## Data Availability

All data are available in the manuscript.

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
