# Peer review of "Chemical Composition, Enantiomeric Distribution, Antimicrobial and Antioxidant Activities of Origanum majorana L. Essential Oil from Nepal"

_molecules, 2022, doi:10.3390/molecules27186136_

Round 1

Reviewer 1 Report

The MS entitled “Chemical Composition, Enantiomeric Distribution, Antimicrobial and Antioxidant Activities of Origanum majorana L. Essential Oil from Nepal” was reviewed very critically. The article is scientifically sound and very critical work has been reported. However, one of the significances of study revolves around the enantiomeric analysis of oils, which, only one sample has been evaluated. Moreover, the article should be corrected for many grammatical and structural mistakes. Some of my suggestions include:

1. Line 18. After “different locations” mention the samples names and codes here. Do not need the separate appearance of such sample codes in parts.

2. line 27. The authors should not state the word “probably” which reflects an uncertainty in MS.  Instead search and ensure that this type of results have not been found anywhere.

3. Line 33. Use proper scientific name of the genus.

4. Line 82. What is meant by aromatic plants?  This may be aroma plants. If aromatic, kindly explain the term.

Line 225. Why only chiral GC-MS for BKT? Why not the authors performed this experiment for the more active sample, KTM? Also, the antimicrobial activities were only performed for KTM. Why not BKT?

5. Line 232. Check the format.

6. Line 263. Correction needed in “majorana can”.

Line 273. Write DPPH in full and then use the abbreviation onward.

Line 278. Remove “the”.

Lines 319-320. Format corrections.

Line 427. How IC50 values were calculated?

Lie 435-436. Revise and rewrite.

Line 438. “bioactive compound” should be “constituents”.

Line 440. “earlier results” should be “previous studies”. Also “against bacteria should be “against the tested bacteria”.

Line 449. Potency instead of potent. These types of grammatical mistakes should be corrected in the MS.  Also rewrite the conclusion.

Line 511. Correct reference 18.

Line 545. Reference 30. Correct India.

Author Response

We are very happy to have such wonderful comments and suggestion to improve the quality of
this manuscript. We have incorporated the missing data as recommended by you. Though the
experiment was already carried, some of these data were not interpreted in systematic way and
some in hurry way to publish it. So, our team realized it very seriously and have tried to add
some results in the manuscript.

Some of the suggestions made by reviewer are answered as:
1. Line 18. After “different locations” mention the samples names and codes here. Do not need the
separate appearance of such sample codes in parts.
Answer: We corrected this part in the manuscript as you suggested.

2. line 27. The authors should not state the word “probably” which reflects an uncertainty in MS.  Instead
search and ensure that this type of results have not been found anywhere.
Answer: Since this type of details study was not found during literature review, except for few particular
components in EO of O. majorana, the term is now corrected to ensure this type results as suggested
you.

3. Line 33. Use proper scientific name of the genus.
Answer: Yes, it is. A proper full name of genus is given now,
4. Line 82. What is meant by aromatic plants?  This may be aroma plants. If aromatic, kindly explain the
term.
Answer: Actually, it refers to the plants having strong fragrance due to volatile compounds. So, it should
be medicinal and aromatic plants in this context as described in the paragraph which is corrected now.

Line 225. Why only chiral GC-MS for BKT? Why not the authors performed this experiment for the more
active sample, KTM? Also, the antimicrobial activities were only performed for KTM. Why not BKT?
Answer: As we mentioned in the earlier paragraph, we now incorporated the missing results for both
samples, chiral GC MS and antimicrobial test. Actually, it happened due to some miscommunication with
our colleagues who carried these analysis.

5. Line 232. Check the format.
Answer: Checked it. For citation, it is in accordance to format.

6. Line 263. Correction needed in “majorana can”.
Answer: Yes, it has been corrected in the manuscript. The sentence is slightly modified.

Line 273. Write DPPH in full and then use the abbreviation onward.
Answer: Yes, it has been corrected as 2,2-diphenyl-1-picrylhydrazyl and followed DPPH subsequently.

Line 278. Remove “the”.
Answer: The manuscript which I have downloaded after revision shows slightly different line number.
Though I have tried to make the correction. Yes, it is also corrected.

Lines 319-320. Format corrections.
Answer: I have used the Mendeley software for citation and followed as accordance to the format of
Journal.

Line 427. How IC 50  values were calculated?
Answer: Using standard protocol, we set the different concentration for EO samples and standard
ascorbic acid in order to lie the absorbance within 0.1 to 0.9 in DPPH Assay method. After this, we took
the triplicate absorbance reading for each concentration of EO samples, control and standard. By using
the formula [Ac-As]/Ac *100, we found the % Scavenging activity for samples and standard. Then a plot
for % inhibition vs., different concentration was prepared. Then using linear curve equation, y= mx+c,
keeping y=50 and slope of that line and intercept, we calculated the ‘X’ concentration giving the IC 50
value. The whole method is clearly explained in manuscript as well. We can elaborate it further if it needs
more. We have repeated this experiment for further confirmation and correction of this data in scientific
way.

Lie 435-436. Revise and rewrite.
Answer: We corrected it now.

Line 438. “bioactive compound” should be “constituents”.
Answer: Yes, it is corrected.

Line 440. “earlier results” should be “previous studies”. Also “against bacteria should be “against the
tested bacteria”.
Answer: Yes, it has been corrected as suggested.
Line 449. Potency instead of potent. These types of grammatical mistakes should be corrected in the
MS.  Also rewrite the conclusion.

Answer: Yes, we corrected it. The grammatical errors were revisited and corrected them properly. The
conclusion is modified a little bit to harmonize well and address the research work after the inclusion of
some results as mentioned above.
Regarding the grammatical correction, we checked it thoroughly to overcome all possible grammatical
mistakes.

Line 511. Correct reference 18.
Answer: This is published book available in the central library of Nepal. So, this is in Nepali date of
publication. If not suitable in format, it can be deleted.

Line 545. Reference 30. Correct India.
Answer: Yes, it is corrected in the manuscript.

Note:
ï‚· All the corrections made for the manuscript as recommended by the reviewer are
highlighted in ‘yellow color’.
ï‚· We have included the agglomerative hierarchical cluster analysis based on the
compositions of EO of marjoram in this manuscript which was another significant part in
this paper.

Thank you once again!

Reviewer 2 Report

Dear Authors,

after reading your manuscript, I have to conclude that it needs to be improved before it can be published and presented to a scientific audience. Here are some of my comments and suggestions on how to improve your article.

Table 1 should be set up differently: choose one parameter for both samples RI or RT. If you choose RI, please also indicate whether it is calculated or derived from the literature. As some compounds were present in both samples, it would be preferable if the table was set up as follows: number of the peak in chromatogram - name of the compound - RI or RT - quantity in sample 1 - quantity in sample 2. This would also make it easier to compare the two samples.

In the discussion part, I would suggest that all the literature data on the composition of O. majorana essential oils should be collated and presented in a table, for example: country-major compound (amount). This would make it much easier to keep track of and compare all the existing data. 

Figure 2 is completely redundant. The compound structures are not difficult for the reader to find in other sources, nor are the compounds rare or newly discovered. This figure can contain the structures of investigated enantiomers.

In Figures 1 and 3, it would be better to write the number of the compound instead of RT at the peak.

Antioxidant activity test is not distinguished in the results and discussion section under a separate title.

At least two assays are commonly used to determine antioxidant activity. Maybe you have the possibility to complement your manuscript with the results obtained using one additional antioxidant activity test (ABTS, ORAC, FRAP). The deferred values of the axes in Figure 5a are questionable. At a concentration of 1000 µg/ml, the IC50 value is not even reached, so there must be some inaccuracy.

I think that the enantiomeric analysis of the compounds should be one of the main strengths of the paper, as the antimicrobial, antifungal and antioxidant properties of O. majorana essential oil with its various compositions are well known and studied. And this should be highlighted in the conclusions.

Author Response

After reading your manuscript, I have to conclude that it needs to be improved before it can be published
and presented to a scientific audience. Here are some of my comments and suggestions on how to
improve your article.
Table 1 should be set up differently: choose one parameter for both samples RI or RT. If you choose RI,
please also indicate whether it is calculated or derived from the literature. As some compounds were
present in both samples, it would be preferable if the table was set up as follows: number of the peak in
chromatogram - name of the compound - RI or RT - quantity in sample 1 - quantity in sample 2. This
would also make it easier to compare the two samples.
Answer: Yes, it is absolutely correct. As you suggested, the table was reset up in accordance to the one
parameter for both samples. Regarding RI, it has been clearly described at bottom of the table. Also, we
have given sample name as S 1 and S 2 representing two locations of Nepal.

In the discussion part, I would suggest that all the literature data on the composition of O.
majorana essential oils should be collated and presented in a table, for example: country-major
compound (amount). This would make it much easier to keep track of and compare all the existing data. 
Answer: We have made some drastic change in its look following your suggestion. We have expressed
the results of previous studies in the table 2 to explicit it for easier comparison after inclusion of few
results.

Figure 2 is completely redundant. The compound structures are not difficult for the reader to find in other
sources, nor are the compounds rare or newly discovered. This figure can contain the structures of
investigated enantiomers.
Answer: Since Figure 2 shows comparison of major chemical compounds present in the two EO samples
along with their structure as discussed in the result and discussion,Therefore, it is thought to include in
the manuscript. This also includes the structure of chiral compound obtained in the both EO of O.
majorana samples like Terpinen-4-ol, cis-Sabinene hydrate, Linalool, Linalyl acetate, Bornyl acetate, α-
Terpineol. β-Caryophyllene. It is mentioned in figure 2 as well.

In Figures 1 and 3, it would be better to write the number of the compound instead of RT at the peak.
Answer: Figure 1 actually shows the GC-MS chromatograms with comparative abundance of each
chemical compounds at different retention time and it illustrates in better way to understand
chromatogram patterns. While, Figure 3 in the manuscript has been corrected.

Antioxidant activity test is not distinguished in the results and discussion section under a separate title.
Answer: Antioxidant activity test is written as separate part in the section 2.6.1 and 2.6.2 of result and
discussion. It is clearly described and interpreted along with DPPH radical scavenging activity in the table
5.

At least two assays are commonly used to determine antioxidant activity. Maybe you have the possibility
to complement your manuscript with the results obtained using one additional antioxidant activity test
(ABTS, ORAC, FRAP). The deferred values of the axes in Figure 5a are questionable. At a concentration
of 1000 µg/ml, the IC 50 value is not even reached, so there must be some inaccuracy.
Answer: Taking it more seriously, we repeated this experiment again along with another assay (FRAP)
and put it more clearly now. During the protocol setting for standard and EOs sample, we took the
different concentration in order to lie the absorbance range of 0.1 to 0.9. Therefore, the axes are looking
different based on our experimental modification to the follow the standard protocol. Actually, EOs have
weaker activity at lower concentration. Therefore, we carried this experiment considering at different
concentration as compared to ascorbic acid as standard.
I think that the enantiomeric analysis of the compounds should be one of the main strengths of the
paper, as the antimicrobial, antifungal and antioxidant properties of O. majorana essential oil with its
various compositions are well known and studied. And this should be highlighted in the conclusions.
Answer: Yes sir, it is now corrected and included the result on highlighting the enantiomeric analysis in
conclusion which was missing.
Note:
ï‚· All the corrections made for the manuscript as recommended by the reviewer are
highlighted in ‘green color’.

ï‚· We have included the agglomerative hierarchical cluster analysis based on the
compositions of EO of marjoram in this manuscript which was significant one in this
paper.

Thank you once again!

Round 2

Reviewer 1 Report

Check for spellings mistakes and grammer.